# Beyond the Spectrum: Unleashing the Potential of Infrared Radiation in Poultry Industry Advancements

**DOI:** 10.3390/ani14101431

**Published:** 2024-05-10

**Authors:** Khawar Hayat, Zunzhong Ye, Hongjian Lin, Jinming Pan

**Affiliations:** 1College of Biosystems Engineering and Food Science, Zhejiang University, Hangzhou 310058, China; 2Key Laboratory of Intelligent Equipment and Robotics for Agriculture of Zhejiang Province, Ministry of Agriculture and Rural Affairs, Hangzhou 310058, China

**Keywords:** infrared, light, radiation, poultry, welfare, farming, production, health

## Abstract

**Simple Summary:**

The poultry industry is advancing through a focus on nutrition, management practices, and technology to enhance productivity by improving feed conversion ratios, disease control, lighting management, and exploring antibiotic; alternatives. Infrared (IR) radiation, with wavelengths ranging from 760 to 10,000 nm, is being used to improve the well-being of humans, animals, and poultry. IR applications like heating, spectroscopy, beak trimming, and other therapies offer health benefits such as improved skin health, wound healing, and enhanced immune responses. In poultry production, IR radiation positively impacts growth performance, gut health, blood profiles, and food safety. Despite its benefits, applications of IR in poultry are limited but show promise for wider adoption. This innovative approach has the potential to revolutionize traditional practices in the industry by optimizing productivity and efficiency while improving animal welfare.

**Abstract:**

The poultry industry is dynamically advancing production by focusing on nutrition, management practices, and technology to enhance productivity by improving feed conversion ratios, disease control, lighting management, and exploring antibiotic alternatives. Infrared (IR) radiation is utilized to improve the well-being of humans, animals, and poultry through various operations. IR radiation occurs via electromagnetic waves with wavelengths ranging from 760 to 10,000 nm. The biological applications of IR radiation are gaining significant attention and its utilization is expanding rapidly across multiple sectors. Various IR applications, such as IR heating, IR spectroscopy, IR thermography, IR beak trimming, and IR in computer vision, have proven to be beneficial in enhancing the well-being of humans, animals, and birds within mechanical systems. IR radiation offers a wide array of health benefits, including improved skin health, therapeutic effects, anticancer properties, wound healing capabilities, enhanced digestive and endothelial function, and improved mitochondrial function and gene expression. In the realm of poultry production, IR radiation has demonstrated numerous positive impacts, including enhanced growth performance, gut health, blood profiles, immunological response, food safety measures, economic advantages, the mitigation of hazardous gases, and improved heating systems. Despite the exceptional benefits of IR radiation, its applications in poultry production are still limited. This comprehensive review provides compelling evidence supporting the advantages of IR radiation and advocates for its wider adoption in poultry production practices.

## 1. Introduction

Recently, there has been a strong emphasis on advancing poultry production through improvements in areas like nutrition, management practices, and technology. Experts in the poultry industry are working on implementing strategies to enhance productivity. These strategies include improving feed conversion ratios, monitoring, and controlling infectious diseases, expanding veterinary services, developing vaccines, optimizing lighting management, enhancing surveillance practices, and exploring natural alternatives to antibiotics to boost growth performance. By focusing on these aspects, the poultry industry aims to achieve significant progress and efficiency in production [1]. In the field of biology, infrared (IR) light is utilized for various operations to enhance the well-being of humans, animals, and poultry. IR radiation consists of electromagnetic radiation with wavelengths longer than those of visible light (750 nm–100 μm), frequencies ranging from 400 THz to 3 THz, and photon energies ranging from 12.4 meV to 1.7 eV. It can be categorized into five bands: near-infrared (NIR), short-wavelength infrared (SIR), mid-infrared (MIR), long-wavelength infrared (LIR), and far-infrared (FIR) [2,3]. Different forms of IR radiation and their uses in different fields are shown in Figure 1. In the field of medicine, IR is extensively utilized, and its prevalence is increasing. The therapeutic efficacy of IR is subject to numerous factors, conditions, and variables such as intensity, duration of treatment, repetition rate, pulse, and frequency, and is specifically used in applications like neurological stimulation, wound healing, tissue regeneration, and cancer therapy [4]. It has been proposed for a range of neurostimulation and neuromodulation practices, demonstrating beneficial impacts on nerve cells [2,5]. IR radiation is involved in various biological processes such as stress signaling, metabolism, cytoskeleton organization, homeostasis, cell proliferation, and differentiation [6]. The IR generates reactive oxygen species (ROS) and increases intracellular calcium ions (Ca^2+^). IR changes the water dynamics affecting cell membrane and mitochondrial function, releasing Ca^2+^ into the cytosol. Elevated Ca^2+^ activates enzymes in cellular respiration, producing ATP via the tricarboxylic acid (TCA) cycle and the electron transport chain (ETC) essential for cellular processes, while ROS and Ca^2+^ regulate cellular signaling. (Figure 2) [7]. Furthermore, infrared (IR) light has the potential to activate cells through water absorption, leading to an increase in the temperature of the plasma membrane and alterations in the electrical capacitance of the cell [6,8]. The above-mentioned studies in the literature demonstrate the advantageous impacts of IR light or radiation and its application for overall well-being. We are excited to delve deeper into the positive effects of IR light to encourage its utilization in poultry and animal husbandry, considering poultry as a major contender. Numerous intricate issues confront the poultry business such as rising production costs and scarce resources, worries about food safety and public health, diseases, and the welfare of birds, and shifting customer preferences [9,10]. The poultry business may overcome several obstacles by implementing IR technology. IR heating systems can help with energy conservation and improve the economy. IR thermal imaging is used to detect physiological abnormalities and monitor chicken health and welfare by measuring body temperature [11,12]. Near-infrared spectroscopy has enabled farmers to assess barn thermal environments, ensure feed quality, and improve price management. This technology allows for rapid evaluation of conditions and feed characteristics, helping to optimize livestock well-being, nutrition, and operational efficiency [13]. IR beak trimming will improve welfare and health concerns [14]. These infrared-based technologies provide automated, non-invasive methods to monitor poultry health and production conditions, which can help producers to improve efficiency, animal welfare, and sustainability in the face of the industry’s multifaceted challenges. Near-infrared spectroscopy has assisted farmers in analyzing barn temperature settings, verifying feed quality, and improving pricing management. This technique enables the fast evaluation of feed properties, thereby improving animal health, nutrition, and overall performance [13]. IR beak trimming will promote welfare and address health problems [14]. These IR-based technologies offer automated, non-invasive approaches to monitoring poultry health and production conditions, allowing farmers to increase efficiency, animal welfare, and sustainability. This review will explore the various uses of IR technology and its advantages within the realm of poultry farming.

## 2. Applications of Infrared Radiation

All organisms are subject to solar electromagnetic radiation, which can have varying effects on biological entities from subcellular components to entire organisms [4]. IR radiation, a form of energy with diverse positive impacts has proven effective in treating various conditions. Initially, IR therapy was limited to heated saunas for controlled exposure over specific durations. Technological advancements have refined the delivery of pure IR radiation, with modern IR-emitting devices incorporating nanoparticles for enhanced therapeutic outcomes [6]. In the food industry, IR technology has become a useful and effective tool for a variety of food- and feed-processing processes. IR heating systems have been used to effectively dry, cook, and sterilize meat, dairy, and egg products while retaining significant nutritional and sensory properties. Studies have demonstrated that IR drying can preserve more vitamins and antioxidants in animal-derived foods than traditional hot-air drying [15,16]. Furthermore, IR surface pasteurization was investigated as a method of improving the microbiological safety of beef and poultry products [17].

However, its application in poultry welfare remains underexplored. Exploring the advantages of IR light in poultry farming could revolutionize production practices (Table 1). In the preceding discourse, we have elucidated the advantageous utilization of IR radiation for enhancing various parameters within the realm of poultry production. Through the targeted deployment of IR radiation, notable improvements in productivity, health, and the well-being of poultry populations can be achieved, thereby emphasizing the significance of this technology in modern poultry-farming practices (Figure 3). In the field of poultry farming, IR has been used in the following different operations to optimize poultry farming practices (Figure 4): IR heating systems were used to improve thermal comfort in poultry houses by IR radiation, which also saves energy [18,19]; IR beak trimmers increase bird welfare [20,21,22]; IR thermography using IR radiation is used to determine the accurate surface temperatures of birds to monitor any abnormal conditions [23,24], IR spectroscopy used IR vibrations to evaluate feed quality [25,26]; and IR was used in computer vision technology for monitoring poultry enterprises operations and bird activities [27,28,29]. The uses of IR to improve different parameters in poultry production include improving growth performance, acting as an antitoxin and anti-pathogenic agent, improving quality control and food safety measures, improving precision poultry practices and profits, and boosting immunity [30,31,32,33,34,35].

## 3. Integrating Infrared Technology for Optimizing Poultry Farming

### 3.1. Infrared Heating Systems

In the realm of animal husbandry, IR heating systems have emerged as a prevalent choice for maintaining optimal temperatures within animal and poultry houses during cold weather conditions. These systems play a crucial role in reducing heat loss and are highly efficient for providing supplemental heat to various species of poultry and animals. Studies [18,19] have underscored the efficacy of IR heating systems in enhancing the growth and well-being of poultry. Sklyar and Postnova’s (2018) research highlighted the significant impact of IR heaters in minimizing heat consumption within poultry houses, particularly during the initial stages of broiler growth, thereby replacing conventional energy-intensive electric lauders. Furthermore, Hajrullin and Teregulov (2016) emphasized the use of IR heating to mitigate heat loss in broiler houses through the enhancement of heating devices and systems within poultry facilities. The advantages of IR heaters, as elucidated by Sklyar and Postnova (2018), are multifaceted and include:

Facilitation of microclimatic adjustments within poultry houses to counteract winter cold peaks without compromising air circulation;Preservation of air moisture levels due to the selective heating nature of infrared radiation, which targets surfaces such as birds, litter, and individuals rather than heating the entire air mass;Maintenance of comfortable conditions for birds and humans through the emission of longwave IR radiation at minimal temperatures;Unified integration of IR heaters into existing infrastructure without impeding technological operations;Acceleration of winter warm-up times in poultry houses by up to 75%, optimizing operational efficiency.

### 3.2. Infrared Beak Trimmer

The IR beak trimmer is an advanced beak-trimming device used in poultry. In the past, simple heat blades were used to trim the beaks of poultry birds, but after the introduction of the IR beak trimmer, it was found that using IR beak trimming instead of heat blade trimming has numerous advantages [1]. Hughes (2020) stated that using the IR beak-trimming method had a statistically non-significant effect on beak morphology and the production performance of hens treated by IR beak trimmer when compared with a control group [20]. Furthermore, histological analysis revealed no neuromas in the beak tissue, indicating that the hens were not in constant pain as a result of the IR trimming procedure. Carruthers et al. (2012) stated that, compared to heat blade trimming, IR blade trimming significantly reduces the frequency of beak morphologies such as beaks with a visible crack on the top or bottom (shovel beak) and beaks with a bubble under the tissue of the beak tip (bubble beak) [21]. Depending on the age at treatment and the degree of treatment, heat blade treatment may result in neuroma formation, a cause of chronic pain [22]. It has been proven that by using IR beak trimming to create a shovel beak, birds do not reduce feed intake or body weight [1]. The results suggested that IR beak trimming is much better than heat blade trimming and should be used to promote bird welfare and health concerns (Figure 5).

### 3.3. Infrared Thermography

Infrared thermography (IRT) uses IR radiation to non-invasively measure surface heat. Localized inflammation or a decrease in blood flow can be detected by IRT [38]. IR imaging devices may aid in any temperature-dependent process, and IRT can be applied in many sectors. For more than 50 years, IRT has been used to detect poultry illnesses in more than 30 bird species. The IRT temperatures of the hock, shank, and foot regions were lower in lame broilers than in control broilers, suggesting that IRT may be useful for detecting subclinical bacterial chondronecrosis with osteomyelitis lesions in broiler chickens [39]. IRT monitors metabolic heat generation in domestic hens [23,24], footpad dermatitis, and handling stress [40,41]. IRT has applications in thermal physiology, injury, disease diagnosis, and animal population counting. IRT combined with biophysical modeling provides a powerful tool for estimating metabolic heat loss, physiological states, relative energy costs of behaviors, and environments [42]. IRT also has the potential to measure superficial temperature variations during the physiological phases of livestock and poultry, making it an efficient tool for health assessment. Integrating Internet of Things technology with IRT is recommended as a future direction for autonomous, efficient, and economical temperature measurement systems in the livestock and poultry industry [43]. Thermal comfort is an essential welfare measure for poultry, and excellent thermal comfort enhances the development of chicken genetic potential and productivity. Thermal imaging was used to assess broiler chickens’ thermal comfort, and a comparison of different ventilation settings revealed that a high wind speed is more favorable for heat transmission and provides greater thermal comfort for broilers [44,45]. Applying these activities at the farm level can enhance bird living conditions and, as a result, bird performance.

### 3.4. Infrared Spectroscopy

Infrared spectroscopy (IRS) generates molecular vibrations through exposure to IR light, serving as a non-destructive analytical tool for real-time monitoring and quality control in the food-processing industry. It is widely recognized as a fundamental spectroscopic method for structure elucidation and chemical identification [36]. IRS plays a crucial role in both the qualitative and quantitative analysis of food products. Its applications range from identifying bacterial strains and quantifying contamination levels to assessing sensory attributes and technical characteristics for food authentication and adulteration detection [36]. In the realm of animal production, understanding the nutritional value of feed ingredients and diets is paramount for sustainable practices [46]. Feed assessment stands as a cornerstone of nutritional research innovation, employing various techniques such as chemical analysis, tabulated values, prediction equations, near-infrared (NIR) reflectance spectroscopy, and in vitro methods. Developing reliable and efficient in vitro procedures for evaluating the nutritional composition of feed components or complete diets is essential. The accuracy of these assessment methods is crucial, as inaccuracies can impact bird performance, escalate feed costs, and pose environmental challenges. In this context, IRS emerges as a contemporary and advanced approach for evaluating feed quality in poultry production [37]. By providing precise information on feed quality, IRS offers a more efficient alternative to traditional methods, contributing to improved outcomes in poultry farming. Both NIR and MIR spectral data can be used to analyze food components such as moisture, protein, and fat [47,48]. Chemometric approaches for quantifying food product qualities using spectral data include partial least squares, artificial neural networks, and support vector machines. This quick, easy, non-destructive, and inexpensive spectroscopic method has been widely utilized for internal and external quality assessments of food items, particularly meat and meat products [25,26]. Barbin et al. (2020) have shown that NIR spectroscopy may be an effective method for identifying adulterations in turkey meat without requiring significant physicochemical examination [49]. Similarly, De Marchi et al. (2017) discovered good findings when utilizing NIR spectroscopy to predict sodium content in commercially processed meat products [50].

### 3.5. Potential of Infrared in Computer Vision for Poultry Monitoring

A computer vision system containing of hardware and software components could make a diverse contribution to monitoring poultry businesses (Figure 6) [51]. IR imaging and thermal sensing have significant potential to enhance computer vision-based monitoring of poultry health, behavior, and welfare in commercial farming operations [27,28,29]. Computer vision and machine learning could be used to monitor various aspects of the poultry industry, including bird health management, object detection, semantic segmentation, body temperature monitoring, body weight measurement, image classification, growth monitoring, chicken tracking and counting, gender detection, and behavior analysis [52]. Further elaborating on computer vision technology. A study developed a model structure to identify sick broilers by using deep learning digital image processing [53]. In another study, a deep poultry vocalization network was used for the early detection of Newcastle disease based on bird vocalization. The method combined multi-window spectral subtraction and high pass filtering to reduce the influence of noise [54]. The system could be used to analyze the activity levels of broiler chickens within the pen. This could provide insights into the birds’ behavior and help to identify any abnormal activity patterns that may be indicative of health or welfare issues [55,56]. The implementation of computer vision technology has significantly improved the weighing and processing systems used in the poultry industry [57]. The poultry industry places great importance on automatically detecting, counting, and monitoring to improve productivity and welfare [58]. Precise and automatic counting of birds is essential for the intelligent management of poultry farms, as the difficulty in precise monitoring of large-scale operations lies in the automatic tracking of each bird [59]. To address this challenge, a deep regression network was introduced to track individual poultry birds using computer vision technology [60]. A recent study proposed a novel method for accurately identifying the gender of chickens using an enhanced version of the popular ResNet-50 deep learning algorithm. The researchers made several key modifications to the standard ResNet-50 architecture, including increasing the network depth, experimenting with customized activation functions, incorporating attention mechanisms, and leveraging transfer learning. The enhanced model has demonstrated impressive performance in accurately distinguishing between male and female chickens, significantly outperforming traditional methods. This innovative technique has the potential to revolutionize various applications in poultry farming, hatchery operations, research, and conservation, offering a reliable and scalable solution for efficient chicken gender identification [61,62]. Temperature measurement is crucial in managing commercial poultry houses, as heat stress can significantly impact bird productivity and farm profitability. Poultry birds are sensitive to environmental temperatures that differ from their optimal body temperature range, leading to reduced feed intake, growth rate, and feed conversion efficiency, as well as increased mortality [63]. In commercial houses, a climate control sensor is installed to measure the surrounding temperature, but this may not accurately reflect the actual body temperature f the birds, which can be influenced by factors such as bird density, ventilation, and radiant heat sources. Accurate temperature measurement is essential for effective climate control and the optimization of bird welfare and farm performance. This challenge can be addressed by installing thermal cameras with computer vision to measure birds’ body temperatures as an indicator of their health. This monitoring method, based on infrared thermal imaging, has the advantages of being non-contact and stress-free, making it increasingly valuable in poultry production for tracking body temperature and detecting diseases [64]. By integrating temperature measurement and computer vision techniques, poultry producers can gain a more comprehensive understanding of their flock’s well-being and optimize their climate control systems to enhance productivity and profitability.

## 4. Applications of Infrared in Poultry Production

### 4.1. Infrared Treatment to Improve Growth Performance

Growth performance is a basic parameter for evaluating the overall health and productivity of a bird. Studies have indicated that variations in lighting parameters such as intensity, source, duration, and color can potentially impact physiological responses, the neuroendocrine system, and, ultimately, the growth performance of broiler chickens [65,66,67,68,69]. Broilers are less stressed when allowed to regulate their own photo period [70]. The wavelength of light has various effects on performance and behavior, and restricted lightening improves the body weight, FCR and immune status of poultry [71]. Exposure to far infrared (FIR) radiation holds promise for enhancing broiler performance by increasing protein metabolism and optimizing environmental conditions by decreasing nitrogen secretion [72]. In another study of broilers, broiler feed efficiency increased in response to exposure to LED light with far-infrared radiation, leading to better growth performance than in other birds reared without FIR exposure. Furthermore, exposure to FIR radiation increases the production efficiency and enhances the physiology of broilers, as indicated by decreased morbidity and mortality [30]. These findings indicate the beneficial effects of IR radiation on improving the overall health and performance of birds (Figure 7). Further studies need to be conducted to determine the specific effect of IR on promoting bird and animal performance.

### 4.2. Infrared Technology in the Poultry Industry to Ensure Food Safety

Chicken meat is a widely consumed and economically accessible source of protein in human diets, with global consumption at significant levels. The need for quality-control measures in meat production is intensifying. In this context, near-infrared spectroscopy (NIRS) emerges as a valuable tool for rapid, nondestructive, and precise analysis of chemical composition [31]. Its application extends to the prediction and evaluation of food and feed quality, indicating its versatility and reliability [32,33,34,35]. Extensive research has demonstrated the efficacy of NIRS in predicting protein content across various meat sources such as pork, beef, commercial broilers, eggs, and even insects. Furthermore, studies have shown that NIRS has the potential to classify chicken breast fillets into distinct quality grades for efficient processing and quality control at the industrial level [72,73,74]. Woody breast (hardness or toughness in breast) conditions in the poultry meat industry is a major concern due to their impact on meat quality, processing efficiency and consumer preferences [75]. Continuous monitoring and prompt identification of the occurrence of woody breast are critically important for ensuring food safety in the poultry industry. A near-infrared (NIR) system was proven to detect and distinguish woody breast tissue from normal breast tissue in chicken fillets [76,77]. Adulterants from the polluted environment in poultry meat pose potential health risks, giving rise to food safety concerns for the entire industry. To identify adulterants in chicken meat, a fusion model was developed using visible–near-infrared and shortwave-infrared hyperspectral imaging to detect abnormalities in chicken breast tissue [78]. These studies provide a baseline for optimizing the use of infrared radiation and exhibit the potential to enhance food safety and quality assurance in commercial poultry processing.

### 4.3. Infrared for Precision Poultry Practices and Economic Gains

The identification of subclinical infections in the poultry industry by using IR technology can help prevent widespread outbreaks and their consequences, such as decreased output and profitability. Researchers developed an automated system to identify sick chickens through the combination of audio processing, thermal imaging, and machine learning techniques to prevent disease spread in poultry farming [79]. This approach can mitigate the economic losses of both producers and consumers of poultry meat. IR radiation, specifically FIR radiation, has been extensively employed in medical sectors for various operations, notably in the treatment of chronic disorders [6]. One study investigated the occurrence of white striping and wooden breast myopathies in chicken breasts and concluded that NIRS could provide a valuable, nondestructive, portable alternative to traditional inspection methods [79]. It has potential applications for efficient identification in industrial processing. Another study indicated that IRT temperatures of the hock, shank, and foot regions were lower in lame broilers, suggesting that IRT has the potential to detect subclinical bacterial chondronecrosis with osteomyelitis lesions in broiler chickens [39]. Studies have shown that inadequate feather coverage reduces egg production and lowers feed conversion efficiency by consuming 18 g of extra feed/bird/day [80]. Recently, a multimodal system integrating color, depth, and thermal infrared imaging was developed to evaluate feather damage in the poultry industry [81]. This system offered a more precise, automated, and economical solution than conventional subjective methods. The estimation of embryonic growth before hatching is a critical challenge for poultry farmers and other stakeholders worldwide. The use of near infrared sensors for monitoring the embryonic development and kinetics of chick embryo somatic and cardiac motions was evaluated [82,83]. Visible–near-infrared spectroscopy is used to quantify yolk content in chicken eggs and influences both the hatching time and the sex of broiler chickens [84]. Near-infrared hyperspectral imaging is used for detecting fertility and early embryonic progression within avian eggs [85]. The detection of male and female chick embryos before hatching via the use of a near infrared sensor for measuring the mobility of embryos represents a breakthrough in the poultry industry [83]. These findings could lead to a new era in sustainable and precision poultry production by optimizing hatchery practices, ensuring chick health, reducing mortality rates, minimizing energy consumption, and enhancing production performance, ultimately mitigating economic losses in the industry.

### 4.4. Infrared as Antitoxins and Reducing Noxious Gases

Toxins in feed are a major problem in the poultry industry and can cause enormous losses to poultry growers. The concentrations of plasma, blood aspartate aminotransferase (AST), alanine aminotransferase (ALT), and albumin are in vivo metabolic parameters that can be checked to evaluate the degree of damage to the liver and kidneys due to increased acid or toxins. Son (2015) reported that using FIR radiation with other light sources was helpful for keeping plasma, AST, and albumin concentrations in blood within the normal range. This can indicate that IR reacts as an antitoxin in the bird’s body [30]. Furthermore, in poultry houses, the control of noxious gases is very important because they can also cause health issues for workers who work inside the house along with birds, especially in broiler houses where litter is added just at the start of the flock and birds grow on it throughout the production period, and is only removed when there is a new flock. Prolonged use of the same litter in chicken houses can lead to the generation of harmful gases, including ammonia (NH3) and carbon dioxide (CO_2_). If the concentration of ammonia in the air surpasses 30 parts per million (ppm), it can have detrimental effects on the respiratory health of chickens. This highlights the critical importance of maintaining proper litter management practices to safeguard the well-being of poultry flocks. Furthermore, the effects of elevated gas levels extend beyond health concerns, impacting the overall productivity of the flock. Under such conditions, both the health and production efficiency of chickens can be significantly compromised. Therefore, effective strategies must be implemented to mitigate the accumulation of these noxious gases within broiler houses. In this context, innovative approaches, such as utilizing IR radiation in combination with conventional lighting systems, have emerged as a promising solution for reducing toxic gas levels in poultry farms [30]. Research by Son (2015) demonstrated the efficacy of this combined approach in enhancing air quality and creating a healthier environment for broiler chickens. By leveraging technological advancements like IR radiation, poultry farmers can proactively address the challenges associated with gas accumulation, thereby promoting better welfare and performance outcomes for their flocks. Furthermore, FIR decreased the number of pathogenic bacteria (*E. coli* and *Salmonella*) on poultry farm floors, and the combination of LED and FIR decreased the concentrations of serum cholesterol and triglyceride in hens without affecting egg quality or layer performance [86]. From the above findings, it can be assumed that IR can be useful for reducing toxic gases and pathogenic microbes in poultry houses in several ways, including through altering bacterial toxins, exerting direct antimicrobial actions, and interfering with microbial activities. While the precise mechanisms require further investigation, IR radiation’s capacity to target several features of these hazardous chemicals makes it a prospective mitigation technique.

### 4.5. Infrared as an Immunity Booster in Poultry

The gut microbiota is a dynamic community of microorganisms residing in the gastrointestinal tract that exerts a significant influence on host health. It is involved in vital functions such as nutrient metabolism, immune modulation, and protection against pathogens. Any disturbance in the balance of this system can have profound implications for overall health [87]. FIR has garnered attention for its potential therapeutic effects on various biological systems. In the context of poultry health, FIR exposure has shown promise in modulating the immune response. Son (2015) demonstrated that exposure to both white and green light combined with FIR radiation can increase levels of immune globulins (IgA, IgG, IgM) in broilers, suggesting a potential mechanism through which FIR influences immune function [30]. The immune system serves as the body’s primary defense mechanism against invading pathogens. By enhancing immune globulin levels, FIR exposure can strengthen the immune response in poultry. This augmentation in immunity can help to combat antigens that threaten the body, thereby promoting overall health and well-being. The modulation of the gut microbiota by FIR may also contribute to these immunomodulatory effects. FIR radiation holds significant promise as a therapeutic tool for enhancing the immune response in poultry by influencing the gut microbiota and immune globulin levels. These outcomes indicate the importance of the interplay between gut microbiota, immune response, and IR exposure for optimizing poultry health outcomes. However, further research into the underlying mechanisms of these effects is needed to fully exploit the therapeutic potential of FIR in managing poultry health.

## 5. Conclusions and Future Perspectives

IR may have health-promoting and a variety of therapeutic effects on cells and tissues, making it more versatile than other electromagnetic wavelengths such as visible light. In recent years, a growing number of new findings have shown that various applications of IR have obvious therapeutic advantages, and the mechanism of IR is becoming better-understood. In this review, we summarized the current literature on IR and its specific uses in the field of poultry production. IR applications include IR heating systems (heating poultry houses by IR radiation saves energy), IR beak trimmers (using an IR beak trimmer increases bird welfare), IR thermography (using IR radiation to find an accurate surface temperature), IR spectroscopy (IR vibrations to evaluate feed quality) and IR in computer vision for poultry monitoring (for monitoring poultry enterprises and birds’ performance). The uses of IR to improve different parameters in poultry production include improving growth performance, acting as an antitoxin and anti-pathogenic agent, improving quality-control and food safety measures, improving precision poultry practices and profit, and boosting immunity. The studies included in the present review indicate that IR technology has promising applications in the poultry industry, ranging from improving growth performance to ensuring food safety and precision poultry practices. Studies have highlighted the efficacy of IR heating systems in enhancing growth and well-being, the advantages of IR beak trimming over traditional methods, and the potential of infrared thermography for health assessment. Additionally, infrared spectroscopy has emerged as a valuable tool for evaluating feed quality. IR technology has been instrumental in reducing toxic gases, pathogens, and in improving immune responses in poultry. Despite the various advantageous applications of IR radiation in poultry farming, its utilization remains relatively restricted. Therefore, it is reasonable to infer that expanding the use of IR in poultry production could yield further beneficial outcomes, potentially enhancing the industry and leading to the availability of safer, purer, and more cost-effective food for consumers. Therefore, further research is needed to determine the beneficial effect of IR technology on promoting poultry production.

## 6. Limitations

While the benefits of IR technology in poultry production are evident, further research is needed to fully understand its potential as an antioxidant, antimicrobial agent, and immune booster within the industry. Additionally, the practical implementation and cost-effectiveness of IR systems on a larger scale need to be thoroughly evaluated to ensure their widespread adoption and sustainability in poultry-farming practices.

## Figures and Tables

**Figure 1 animals-14-01431-f001:**
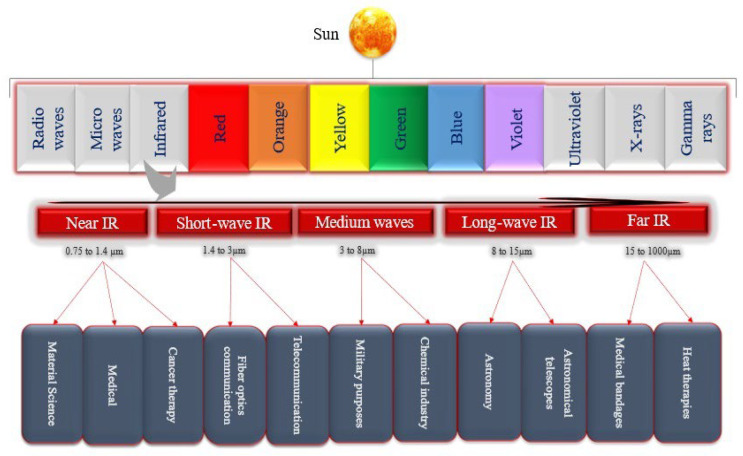
Different forms of infrared radiation and their uses in different sectors.

**Figure 2 animals-14-01431-f002:**
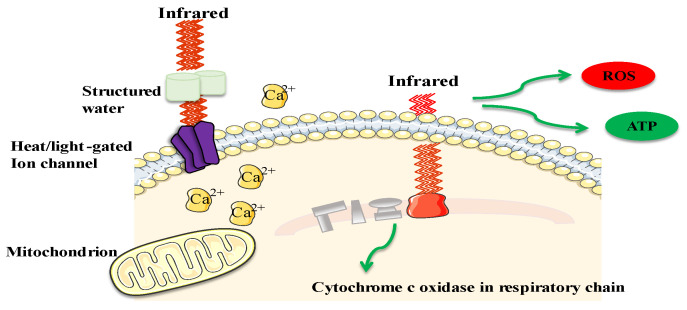
Mechanism of action of IR at the cellular level. IR generates ROS and increases intracellular Ca^2+^. Changes in water dynamics affect membrane and mitochondrial function, releasing Ca^2+^ into the cytosol. Elevated Ca^2+^ activates enzymes in cellular respiration, producing ATP via TCA and ETC, essential for cellular processes, while ROS and Ca^2+^ regulate cellular signaling.

**Figure 3 animals-14-01431-f003:**
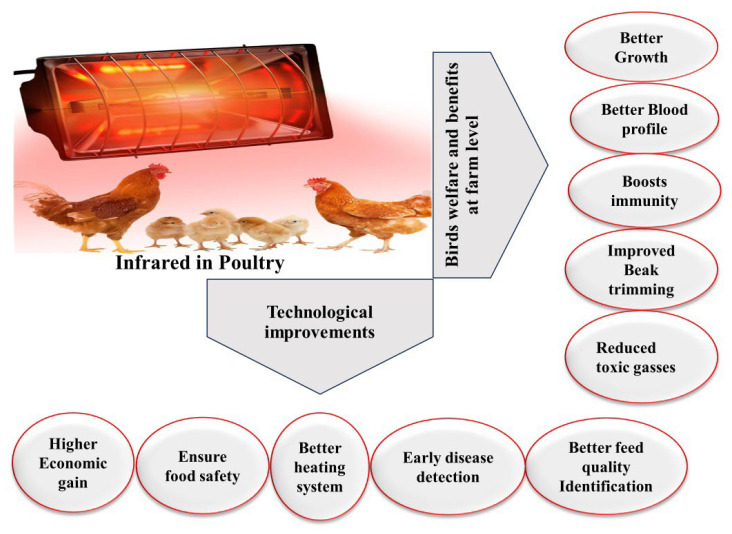
Overall impact of infrared radiation on poultry practices.

**Figure 4 animals-14-01431-f004:**
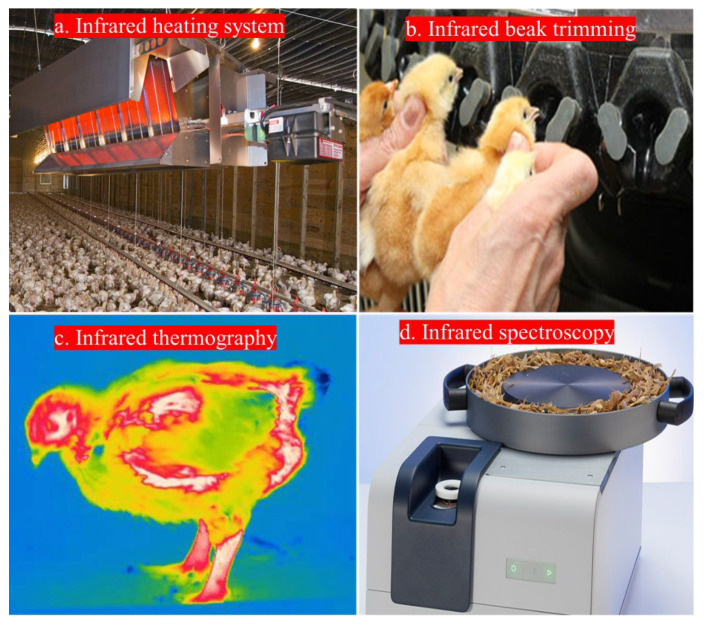
Integrating infrared technology for optimizing poultry farming: (**a**) infrared heating system to improve heating control and preserve energy; (**b**) infrared beak trimming improves bird welfare; (**c**) infrared thermography to detect bird body temperature; and (**d**) infrared spectroscopy to evaluate poultry feed quality.

**Figure 5 animals-14-01431-f005:**
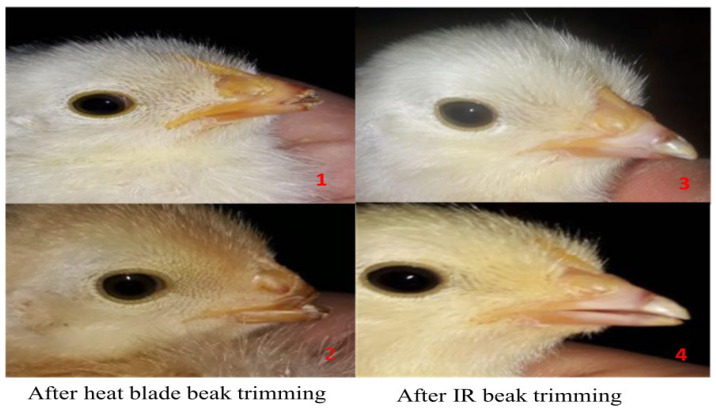
Physical appearance of beak after trimming: (1–2) beak trimming using a heat blade showing crack in beak and abnormal shape; and (3–4) infrared beak trimming showing normal beak appearance.

**Figure 6 animals-14-01431-f006:**
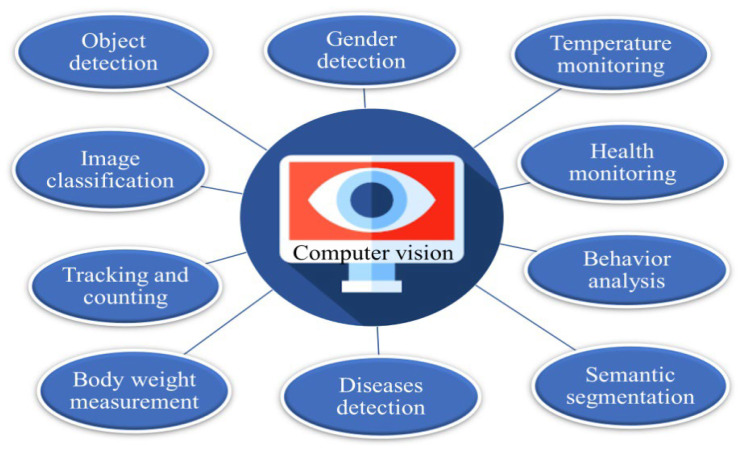
Infrared in computer vision for poultry monitoring.

**Figure 7 animals-14-01431-f007:**
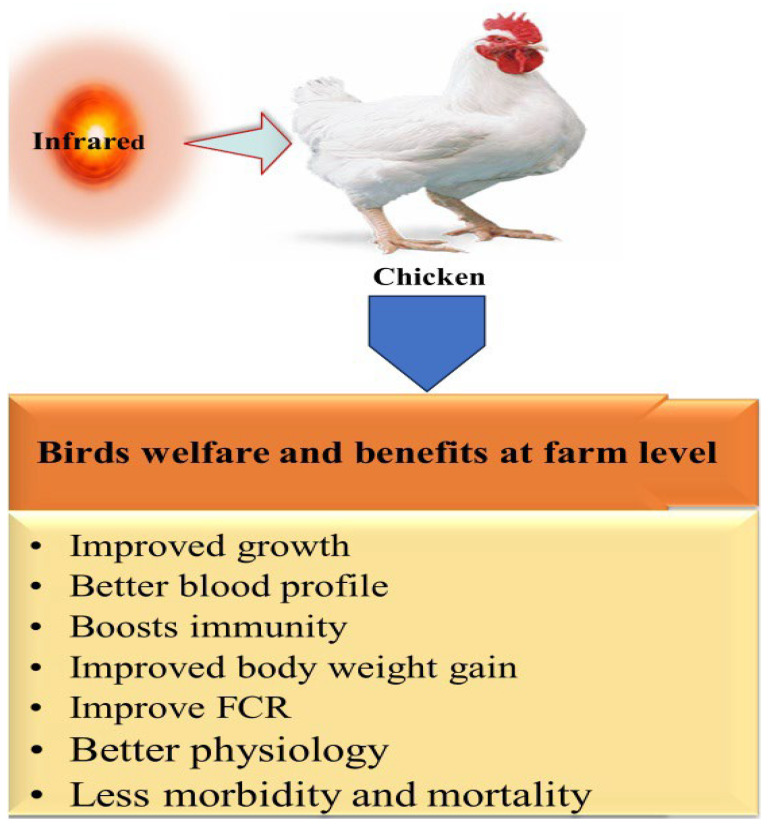
Effect of IR radiation on poultry growth parameters.

**Table 1 animals-14-01431-t001:** Infrared treatment in poultry production.

Parameter	Radiation	Wavelength	Target	Results	Scientists
Feed efficiency	FIR	8–10 μm	Broilers	Improved	[30]
Heating system	IR heaters	5.6–100 μm	Broilers	Improved	[18]
Growth	FIR LED	8–10 μm	Broilers	Improved	[30]
Toxic gases	FIR radiation	8–10 μm	Broilers	Decreased	[30]
Feed Quality	IRS	10–15 μm	Poultry, animals	Improved	[36,37]
Blood profile	FIR	8–10 μm	Broilers	Improved	[30]
Immunity	IR, FIR	8–20 μm	Broilers	Improved	[30]
Beak trimming	IR	8–20 μm	Broilers	Improved	[1,20]
Body weight gain	FIR		Broilers	Improved	[30]
Hydrocarbon	FIR		Broilers	Decreased	[30]

## Data Availability

No new data were created or analyzed in this study. Data sharing is not applicable to this article.

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
