# Peer review of "Beyond the Spectrum: Unleashing the Potential of Infrared Radiation in Poultry Industry Advancements"

_animals, 2024, doi:10.3390/ani14101431_

Round 1
Reviewer 1 Report
Comments and Suggestions for Authors
In poultry, IR enhances growth, gut health, blood health, and food safety, promising to transform traditional practices by increasing productivity and improving animal welfare. However, the review paper could be improved by addressing gaps. It overlooks the potential of infrared in computer vision for poultry monitoring, contains unnecessary chemical details, and lacks sufficient illustrative figures. Simplifying the chemical introduction and incorporating more visuals and information on computer vision applications could make the review more comprehensive and accessible, better highlighting IR light's potential in revolutionizing poultry industry practices.
Author Response
Response to Reviewer 1:
Problem: In poultry, IR enhances growth, gut health, blood health, and food safety, promising to transform traditional practices by increasing productivity and improving animal welfare. However, the review paper could be improved by addressing gaps. It overlooks the potential of infrared in computer vision for poultry monitoring, contains unnecessary chemical details, and lacks sufficient illustrative figures. Simplifying the chemical introduction and incorporating more visuals and information on computer vision applications could make the review more comprehensive and accessible, better highlighting IR light's potential in revolutionizing poultry industry practices.
Response: Thank you very much for your kind suggestions. We followed your suggestion and added the “potential of infrared in computer vision for poultry monitoring” removed unnecessary details in the introduction, and generated more figures, which can improve the manuscript quality.
We added the following data for potential of infrared in computer vision for poultry monitoring (Line 211-249).
3.5. Potential of infrared in computer vision for poultry monitoring
A computer vision system containing of hardware and software components could make a diverse contribution to monitoring poultry businesses (Fig. 6) [53]. IR imaging and thermal sensing have significant potential to enhance computer vision-based monitoring of poultry health, behavior, and welfare in commercial farming operations [28-30]. Computer vision and machine learning could be used to monitor various aspects of the poultry industry, including bird health management, object detection, semantic segmentation, body temperature monitoring, body weight measurement, image classification, growth monitoring, chicken tracking and counting, gender detection, and behavior analysis [54]. Further elaborating on computer vision technology. A study developed a model structure to identify sick broilers by using deep learning digital image processing [55]. In another study, the deep poultry vocalization network was used for the early detection of Newcastle disease based on bird vocalization. The method combined multi-window spectral subtraction and high-pass filtering to reduce the influence of noise [56]. The system could be used to analyze the activity levels of broiler chickens within the pen. This could provide insights into the birds' behavior and help to identify any abnormal activity patterns that may be indicative of health or welfare issues [57-58]. The implementation of computer vision technology has significantly improved the weighing and processing systems used in the poultry industry [59]. The poultry industry places great importance on automatically detecting, counting, and monitoring to improve productivity and welfare [60]. Precise and automatic counting of birds is essential for the intelligent management of poultry farms, as the difficulty in precise monitoring of large-scale operations lies in the automatic tracking of each bird [61]. To address this challenge, a deep regression network has been introduced to track individual poultry birds using computer vision technology [62]. A recent study has proposed a novel method for accurately identifying the gender of chickens using an enhanced version of the popular ResNet-50 deep learning algorithm. The researchers have made several key modifications to the standard ResNet-50 architecture, including increasing the network depth, experimenting with customized activation functions, incorporating attention mechanisms, and leveraging transfer learning. The enhanced model has demonstrated impressive performance in accurately distinguishing between male and female chickens, significantly outperforming traditional methods. This innovative technique has the potential to revolutionize various applications in poultry farming, hatchery operations, research, and conservation, offering a reliable and scalable solution for efficient chicken gender identification [63–64]. Temperature measurement is crucial in managing commercial poultry houses, as heat stress can significantly impact birds productivity and farm profitability. Poultry birds are sensitive to environmental temperatures that differ from their optimal body temperature range, leading to reduced feed intake, growth rate, and feed conversion efficiency, as well as increased mortality [65]. In commercial houses, a climate control sensor is installed to measure the surrounding temperature, but this may not accurately reflect the actual body temperature f the birds, which can be influenced by factors such as birds density, ventilation, and radiant heat sources. Accurate temperature measurement is essential for effective climate control and the optimization of birds welfare and farm performance. This challenge can be addressed by installing thermal cameras with computer vision to measure birds body temperature as an indicator of their health. This monitoring method based on infrared thermal imaging has the advantages of being non-contact and stress-free, making it increasingly valuable in poultry production for tracking body temperature and detecting diseases [66]. By integrating temperature measurement and computer vision techniques, poultry producers can gain a more comprehensive understanding of their flock's well-being and optimize their climate control systems to enhance productivity and profitability.
Furthermore, we added following new figures
Figure 3. Overall impact of infrared radiation on poultry practices
Figure 4. Integrating infrared technology for optimizing poultry farming a: The Infrared heating system to improve heating control and preserve energy. b: Infrared beak trimming improve birds welfare. c: Infrared thermography to detect bird body temperature. d: Infrared spectroscopy to evaluate poultry feed quality
Figure 5. Physical appearance of beak after trimming; 1-2: Beak trimming using heat blade showing crack in beak and abnormal shape; 3-4 Infrared beak trimming showing normal beak appearance
Figure 6. Infrared in computer vision for poultry monitoring
Figure 7. Effect of IR radiation on poultry growth parameters

Reviewer 2 Report
Comments and Suggestions for Authors
The review is interesting, but it is suggested that the authors consider the following points:
L.24. Change "rays" to "light or radiation"
L.71. Homologize the way in which the word "Figure" is presented. Improve the quality of figure 2. The mechanisms of action are not well appreciated, perhaps it would be good to make a description at the bottom of the figure.
L.92. The resolution of Figure 3 is very low, please improve it.
L.138. Include how much the productive parameters were improved. Place a table, etc.
L.138. Place the micrographs.
L.149. Include images where diseases in birds are diagnosed based on studies that have already been carried out.
L.150. "non-invasively"
L.173. Include how IR studies are carried out to evaluate food quality. Are only the IR spectra compared? What IR region is used (near, middle or far)?
L.196. It would be appropriate to include a comparative table or figure showing the improvement in production parameters when IR light is used.
L.270. Be more explicit in the mechanism of action of IR reducing toxins, bacteria and harmful gases.
L.283. subscript.
L.292. light or radiation
L.313. light or radiation

Author Response
Response to Reviewer 2:
Problem: L.24. Change "rays" to "light or radiation"
Response: Thank you very much for your kind suggestion. We changed according to your suggestion.
Problem: L.71. Homologize the way in which the word "Figure" is presented. Improve the quality of figure 2. The mechanisms of action are not well appreciated, perhaps it would be good to make a description at the bottom of the figure.
Response: Thank you very much for your kind suggestion. We changed according to your suggestion.
Problem: L.92. The resolution of Figure 3 is very low, please improve it.
Response: Thank you very much for your kind suggestion. We replaced that figure with a new one.
Problem: L.138. Include how much the productive parameters were improved. Place a table, etc.
Response: Thank you very much for your kind suggestion. We changed according to your suggestion. As beak trimming support to produce negative effect on birds health and performance so in most of the cases birds performance will decreased but here it is found that IR beak trimming will not decrease the performance. We changed the text as follows:
IR beak trimming method had statistically non-significant effect on beak morphology and the production performance of hens treated by IR beak trimmer when compared with control group
Problem: L.138. Place the micrographs.
Response: Thank you very much for your kind suggestion. There is no micrograph available for beak trimming; however, we generated a figure for IR beak trimming and heat blade beak trimming.
Figure 5. Physical appearance of beak after trimming; 1-2: Beak trimming using heat blade showing crack in beak and abnormal shape; 3-4 Infrared beak trimming showing normal beak appearance.
Problem: L.149. Include images where diseases in birds are diagnosed based on studies that have already been carried out.
Response: Thank you very much for your kind suggestion. We generated a figure for IR beak trimming and heat blade beak trimming, presenting harmful effects of heat blad beak trimming mentioned above.
Problem: L.150. "non-invasively"
Response: Thank you very much for your kind suggestion. We changed according to your suggestion.
Problem: L.173. Include how IR studies are carried out to evaluate food quality. Are only the IR spectra compared? What IR region is used (near, middle or far)?
Response: Thank you very much for your kind suggestion. We changed according to your suggestion. We added the data as follow:
Both NIR and MIR spectral data can be used to analyze food components such as moisture, protein, and fat [49-50]. Chemometric approaches for quantifying food product qualities using spectral data include partial least squares, artificial neural networks, and support vector machines. This quick, easy, non-destructive, and inexpensive spectroscopic method has been widely utilized for internal and external quality assessments of food items, particularly meat and meat products [26-27]. Barbin et al. (2020) have shown that NIR spectroscopy may be an effective method for identifying adulterations in turkey meat without requiring significant physicochemical examination [51]. Similarly, De Marchi et al. (2017) discovered good findings when utilizing NIR spectroscopy to predict sodium content in commercially processed meat products [52](line 202-209)
Problem: L.196. It would be appropriate to include a comparative table or figure showing the improvement in production parameters when IR light is used.
Response: Thank you very much for your kind suggestion. We changed according to your suggestion. We generated a figure for improvement in production parameters when IR light is used.
Figure 7. Effect of IR radiation on poultry growth parameters
Figure 3. Overall impact of infrared radiation on poultry practices
Problem: L.270. Be more explicit in the mechanism of action of IR reducing toxins, bacteria and harmful gases.
Response: Thank you very much for your kind suggestion. We changed according to your suggestion. We added the data as follow:
From the above findings, it can be assumed that IR can be useful for reducing toxic gases and pathogenic microbes, in poultry houses through potential ways include altering bacterial toxins, exerting direct antimicrobial actions, and interfering with microbial activities. While the precise mechanisms require further investigation, IR radiation's capacity to target several features of these hazardous chemicals makes it a prospective mitigation technique (line 337-342).
Problem: L.283. subscript.
Response: Thank you very much for your kind suggestion. We changed according to your suggestion.
Problem: L.292. light or radiation
Response: Thank you very much for your kind suggestion. We changed according to your suggestion.
Problem: L.313. light or radiation
Response: Thank you very much for your kind suggestion. We changed according to your suggestion.

Reviewer 3 Report
Comments and Suggestions for Authors
The review article “Beyond the Spectrum: Unleashing the Potential of Infrared Radiation in Poultry Industry Advancements” provides an overview of the potential applications of infrared (IR) radiation in poultry production and highlights its benefits for both animals and humans.
The review is concise, well-written, and presents valuable information on several promising aspects of IR technology in improving various aspects of poultry farming. The manuscript can be accepted after minor revision.
- Although the introduction briefly mentions strategies that aim to improve productivity in poultry production, such as feed conversion ratios, disease control, and antibiotic alternatives, it lacks contextualization or background information on the current challenges facing the poultry industry. I suggest adding a brief overview of the key challenges and trends in poultry production, that would provide a better understanding of the relevance and importance of integrating IR technology into poultry farming practices.
- The images presented in Figure 3 are not clear.
- The section on the applications of IR radiation should be expanded to include a discussion of the application of IR in food processing and also the impact of IR radiation on food quality.
- L 185-189: “While in vivo studies involving live animals offer valuable insights into nutrient utilization and bird performance prediction, they are resource-intensive and time-consuming. In contrast, developing reliable and efficient in vitro procedures for evaluating the nutritional composition of feed components or complete diets is essential.”
Please take into consideration that in vitro procedures cannot be compared with in vivo animal trials when you talk about nutrient utilization (nutrient digestibility, absorption…). Moreover, IRS can serve as an advanced approach for evaluating feed quality and can be compared only with the traditional or the reference methods employed for the feed quality assessment. IRS cannot provide data regarding the nutrients' digestibility.
I suggest you correct this paragraph.
Author Response
Response to Reviewer 3:
Problem: Although the introduction briefly mentions strategies that aim to improve productivity in poultry production, such as feed conversion ratios, disease control, and antibiotic alternatives, it lacks contextualization or background information on the current challenges facing the poultry industry. I suggest adding a brief overview of the key challenges and trends in poultry production, that would provide a better understanding of the relevance and importance of integrating IR technology into poultry farming practices.
Response: Thank you very much for your kind suggestion. We changed according to your suggestion. We added the data as follow:
Numerous intricate issues confront the poultry business, such as rising production costs and scarce resources, worries about food safety and public health, diseases and welfare of birds, and shifting customer preferences [10-11]. The poultry business may overcome several obstacles by implementing IR technology. IR heating systems can help with energy conservation and improve the economy. IR thermal imaging is used to detect physiological abnormalities and monitor chicken health and welfare by measuring body temperature [12–13]. Near-infrared spectroscopy has enabled farmers to assess barn thermal environments, ensure feed quality, and improve price management. This technology allows for rapid evaluation of conditions and feed characteristics, helping optimize livestock well-being, nutrition, and operational efficiency. [14]. IR beak trimming will improve welfare and health concerns [15]. These infrared-based technologies provide automated, non-invasive methods to monitor poultry health and production conditions, which can help producers improve efficiency, animal welfare, and sustainability in the face of the industry's multifaceted challenges. Near-infrared spectroscopy has assisted farmers in analyzing barn temperature settings, verifying feed quality, and improving pricing management. This technique enables fast evaluation of feed properties, thereby improving animal health, nutrition, and overall performance. [14]. IR beak trimming will promote welfare and address health problems [15]. These IR-based technologies offer automated, non-invasive approaches to monitoring poultry health and production conditions, allowing farmers to increase efficiency, animal welfare, and sustainability (Line 72-88).
Problem: The images presented in Figure 3 are not clear.
Response: Thank you very much for your kind suggestion. We changed figure 3 with better suited figure to the manuscript.
Problem: The section on the applications of IR radiation should be expanded to include a discussion of the application of IR in food processing and also the impact of IR radiation on food quality.
Response: Thank you very much for your kind suggestion. We changed according to your suggestion. We added the data as follow:
In the food industry IR technology has developed as a useful and effective tool for a variety of food and feed processing processes. IR heating systems have been used to effectively dry, cook, and sterilize meat, dairy, and egg products while retaining significant nutritional and sensory properties. Studies have demonstrated that IR drying can preserve more vitamins and antioxidants in animal-derived foods than traditional hot air drying [16-17]. Furthermore, IR surface pasteurization has been investigated as a method of improving the microbiological safety of beef and poultry products [18] (line 101-106).
And
In the fiels of the poultry IR have been used in different operations to optimizing poultry forming (Fig. 4) include IR heating systems to improve thermal comforts in poultry houses by IR radiation which also saves the energy [19-20], IR beak trimmers increases bird welfare [21-23], IR thermography using IR radiation to find accurate surface temperature of birds to monitor ant abnormal condition [24-25], IR spectroscopy uses IR vibration to evaluate feed quality [26-27] and IR in computer vision for monitoring poultry enterprises operations and birds activities [28-30]. The uses of IR to improve different parameters in poultry production include improving growth performance, acting as an antitoxin and anti-pathogenic agent, improving quality control and food safety measures, improving precision poultry practices and profit and boosting immunity [31-36] (line 112-119).
Problem: L 185-189: “While in vivo studies involving live animals offer valuable insights into nutrient utilization and bird performance prediction, they are resource-intensive and time-consuming. In contrast, developing reliable and efficient in vitro procedures for evaluating the nutritional composition of feed components or complete diets is essential.”
Please take into consideration that in vitro procedures cannot be compared with in vivo animal trials when you talk about nutrient utilization (nutrient digestibility, absorption…). Moreover, IRS can serve as an advanced approach for evaluating feed quality and can be compared only with the traditional or the reference methods employed for the feed quality assessment. IRS cannot provide data regarding the nutrients' digestibility.
I suggest you correct this paragraph.
Response: Thank you very much for your kind suggestion. We corrected according to your suggestion. We deleted the controversial statement (line 195-197).

Round 2
Reviewer 1 Report
Comments and Suggestions for Authors
After carefully review all the content, since the author has addressed all the comments, the paper is ready to be published.